# Application of the 5th WHO Guidelines for the Diagnosis of Lung Carcinoma in Small Lung Biopsies in a Tertiary Care Center: Is Insecurity of Pathologists for the Accurate Diagnosis Justified?

**DOI:** 10.3390/diagnostics14182090

**Published:** 2024-09-21

**Authors:** Manuela Beckert, Christian Meyer, Thomas Papadopoulos, Georgia Levidou

**Affiliations:** Department of Pathology, Nuremberg Clinic, Paracelsus Medical University, 90419 Nuremberg, Germany; christian.meyer@klinikum-nuernberg.de (C.M.); thomas.papadopoulos@klinikum-nuernberg.de (T.P.)

**Keywords:** lung carcinoma, small lung biopsies, p40 versus p63, PD-L1, histological diagnosis

## Abstract

Background/Objectives: The diagnosis of lung carcinoma (LC) is currently performed in small biopsies and according to the WHO classification by using limited stains to spare tissue for molecular testing. This procedure, however, often causes diagnostic uncertainty among pathologists. Methods: In this retrospective analysis, we compared the diagnosis made by these guidelines in 288 lung biopsies with that using more stains, as retrieved from our archive. We also compared the results of p63 and p40 immunoexpression and investigated the diagnostic role of p53/Rb1. Results: In our investigation, we reached a definite diagnosis with a mean number of one stain compared with six stains in the original diagnostic procedure, with a 97.3% concordance rate. Only in the case of metastases, a clear advantage is proven in the use of more stains, especially in the absence of clinical information. We also found a comparable utility of p40 and p63 for the diagnosis of squamous cell carcinoma, despite the higher p63 expression in other histological types. Moreover, normal p53/Rb1 expression could be utilized for the exclusion of small-cell LC. Conclusions: Our study confirms the diagnostic certainty achieved by the suggestions of the WHO classification and justifies the potential insecurity in the absence of adequate communication with the treating clinician.

## 1. Introduction

Lung carcinoma (LC), which represents the majority of all malignant lung tumors, is the fourth most common cancer diagnosis in Europe and the number one death cause [1]. Since a couple of years, the most relevant subclassification of lung carcinoma has been the distinction between small-cell lung cancer (SCLC) and non-small-cell lung cancer (NSCLS). Nowadays, a precise subclassification of NSCLC into squamous cell carcinoma (SCC), different types of adenocarcinoma (ADC), and large-cell carcinoma is indispensable and forms the basis for further therapeutic decisions. Even ancillary molecular testing is based on this subclassification [2,3].

Biopsy specimens submitted to pathological institutes have become increasingly smaller, mainly due to the development of less invasive biopsy techniques like endobronchial or needle aspiration biopsy [3]. In addition, 70% of lung carcinoma patients present with advanced-stage disease, rendering a resection with acquisition of extended tumor material not possible [4,5]. These factors lead to the predominance of very small biopsies for the daily diagnostic routine. Every diagnostic procedure, from classical HE morphological stain to distinct molecular testing, and subsequently, every therapeutic decision, is based on these small biopsy specimens. Pathologists are confronted with this challenge when diagnosing lung cancer. Therefore, there is a necessity for an efficient diagnostic algorithm that would on the one hand secure accurate diagnosis while sparing tissue for further molecular testing on the other.

According to the latest WHO classification for thoracic tumors [2], the diagnosis in small lung biopsies should be made by morphology if possible. If there is a clear morphological squamous cell or adenocarcinoma pattern, the diagnosis of SCC or ADC should be performed without further immunohistochemical techniques. In poorly differentiated tumors, the immunohistochemical staining for p40 should be used to detect a potential squamous differentiation, whereas PAS-stain (for the detection of mucin) and/or TTF-1 positivity suggests the diagnosis of an ADC. If neither p40 nor PAS or TTF-1 positive reaction is present, the diagnosis of Non-Small-Cell Carcinoma Not Otherwise Specified (NSCLC NOS) should be made. Furthermore, testing for neuroendocrine markers should only be performed in cases with a possible neuroendocrine morphology. Metastatic carcinoma should be excluded by using the clinical information, and further testing with the use of transcription factors in cases of TTF-1 negative ADC should be avoided in the absence of clinical evidence [2].

Nowadays, with the broad use of immune checkpoint inhibitors for cancer treatment, the PD-L1-status (programmed death-ligand 1-status) opens up a further therapeutic option in cases of NSCLC. In general, treatment with immune checkpoint inhibitors is based on a positive tumor proportion score (TPS), which means that tumor cells show a positive reaction [6]. In addition, an IC (immune cells) score with recognition of an >10% area occupied by PD-L1-positive immune cells (lymphocytes, dendritic cells, macrophages, and granulocytes) in the whole tumor area is also known as a predictive marker of tumor response to immune checkpoint-inhibitor therapy [6,7,8].

In this retrospective analysis, we attempt to compare the diagnostic certainty of using the WHO-proposed diagnostic procedure with a limited panel of ancillary testing to our former routine using extended testing to rule out metastases and morphological unexpected features. Additionally, we compared the p40 immunohistochemical stain for the detection of squamous differentiation suggested by the latest WHO classification with the well-known squamous marker p63. The potential diagnostic utility of aberrant p53/Rb1 expression in the diagnosis of SCLC is investigated. Finally, PD-L1 positivity according to TPS and IC-score is also evaluated.

## 2. Materials and Methods

### 2.1. Sample Collection

This is a retrospective study on archival material from small lung biopsies diagnosed in the Department of Pathology, Klinikum, Nuremberg, between 1 September 2020 and 31 May 2021. Our goal was to recruit approximately 300 small lung biopsies. The archival material was collected using the respective codes for lung biopsies having a malignant result in the Nexus system used in our department for the histological reports. For this purpose, lung biopsies with the diagnosis of lung carcinoma are coded as “LUB7”, whereas lung biopsies with a metastasis are coded as “LUB12”. Through this procedure, 303 lung biopsies were found. Fifteen cases had to be excluded from the present investigation, since the archival material was no longer in our department during the study. Finally, 288 cases are included in the present study, consisting of 242 primary lung carcinomas and 46 metastases.

In each case, the histological diagnosis and the number of performed staining procedures (histochemical and immunohistochemical) were documented. Relevant clinical information, such as age, gender, biopsy technique, or tumor location, was also obtained. Moreover, information regarding the number and type of clinical procedures performed to obtain material for the histological diagnosis was collected. We also documented all the available postbioptical information, for example, the presence of subsequent lung resection or molecular analysis.

### 2.2. Histological Evaluation

In all cases, the hematoxylin and eosin (HE) slides were reviewed by two pathologists (GL, MB), one of them with many years of experience in diagnosing lung specimens and blinded regarding the already performed histological diagnosis. During this evaluation, the guidelines proposed by the 5th WHO classification for the diagnosis of lung cancer in small biopsies were used, trying to reach a diagnosis with a minimum number of additional techniques [2]. In this context, cases with classical features of squamous differentiation in the form of keratin pearls, individual cell keratinization, and/or intercellular bridges were defined as SCC, without performing any additional staining. Accordingly, cases with a papillary, acinar, or lepidic growth pattern were diagnosed as ADC on HE grounds. In cases in which an ADC was suspected, a PAS staining was reviewed for the presence of mucin production. If there was no conclusive pattern that could lead to a definite histological diagnosis, immunohistochemical results were evaluated.

### 2.3. Immunohistochemical Analysis

According to the recent WHO lung classification [2], we used only two immunohistochemical markers, namely p40 as a marker of squamous differentiation and thyroid transcription factor (TTF-1) for ADC. In cases with a suspicion of neuroendocrine morphology, an immunohistochemical staining for synaptophysin was additionally evaluated. In all cases with a NSCLC, the PD-L1 immunohistochemical staining was re-evaluated, additionally determining the IC-score. The number of techniques that were used for the diagnosis was documented, and this final diagnosis was compared with the one made with the extensive use of immunohistochemical markers according to our archives. The antibodies employed for this purpose were the following: PD-L1 (clone ZR3, Zeta corporation, Monrovia, CA, USA, 1:100), TTF-1 (clone 8G7G3/1, Agilent, Santa Clara, CA, USA, ready to use), p40 (clone ZR8, Zeta corporation, 1:200).

### 2.4. Immunohistochemical Analysis of p53 and Rb1

In cases with neuroendocrine morphology and immunohistochemical proof of neuroendocrine differentiation through synaptophysin, in which there was doubt regarding the presence of small-cell or large-cell morphology, we also performed immunohistochemical analysis for Rb1 and p53. The following antibodies were used: for Rb1, retinoblastoma clone 1f8 (ready to use), Vitro Master diagnostic, and for p53, p53 protein, clone DO-7 (ready to use), Agilent.

### 2.5. Immunohistochemical Analysis of p40 versus p63

In order to compare the results obtained from the two known markers used according to the literature for the recognition of squamous differentiation, we evaluated the immunoreactivity against p40 and p63 in all cases. p63 immunohistochemical analysis was performed using the p63 protein antibody, clone DAK-p63 (ready to use), Agilent. Both antibodies were evaluated on the basis of the percentage of positive tumor cells, as follows: no staining; 1–10%, low expression; 10–50%, moderate expression; 50–80%, extensive expression.

### 2.6. Statistical Analysis

Numerical parameters are reported as mean, ±standard deviation (SD), and range. Categorical variables are present in the form of percentages and/or number of cases. Comparisons of the parameters were made using parametrical (Student *t*-test, paired *t*-test, Chi-square test), as appropriate. The canonical distribution of our data was evaluated with the Kolmogorov–Smirnov test. The statistical and descriptive analysis was performed using the statistical package STATA 11.0/SE for Windows. A *p*-value of <0.05 was considered statistically significant.

## 3. Results

### 3.1. Patient Collective

A total of 288 cases were included in the present investigation, consisting of 175 male and 113 female patients (male-to-female ratio: 1.54). The male-to-female ratio among patients with a primary lung cancer was comparable, namely 1.52 (146/96).

Patients’ age ranged from 34 to 95 years with a SD ± 9.8. The mean age of patients with lung carcinoma was 68 years (range 50–90, SD ± 9.9) in male and 70 (range 39–86, SD ± 9.3) in female patients. The mean age for patients whose final diagnosis was ADC was 69 years (range 51–86, SD ± 8.8), for patients with SCC, it was 70 years (range 50–90, SD ± 9.4), and for those with SCLC, it was 68 years (range 55–83, SD ± 7.4). The mean age of patients with a metastatic disease was 69 (range 34–95, SD ± 10.6). Patients’ demographic characteristics are shown in Table 1.

### 3.2. Type of Bioptical Material

The bioptical material, which was used for the histological diagnosis, consisted of 171 cases (59.3%) of material gained through bronchoscopy, in 27 (9.3%) cases of endobronchial ultrasound-guided transbronchial needle aspiration (EBUS TBNA), in 50 (17.4%) cases of CT-guided peripheral lung biopsy, and in the remaining 40 (13.9%) cases, of an ultrasound-guided peripheral lung biopsy. In 163 patients, we also received cytological material from bronchoalveolar lavage (BAL), which was proven to be positive in 43 cases (26.4%) and suspicious in one case (0.6%). In 60 cases, we received additional biopsies gained through endoscopy and EBUS-TBNA material from regional lymph nodes. A significant proportion of these cases, namely 68.3% (41/60), showed a lymph node metastasis in this material.

### 3.3. Tumor Location

The tumor localization is presented in Table 2. In 117 cases, the tumor is located in the left lung, and in 122 cases, in the right lung. In three cases, the tumor is located very centrally and cannot be assigned to any lung lobe. In the left lung, 81 cases were located in the central segments of the lung lobes, and 36 cases were in the peripheral lung lobes. In the right lung, 67 cases were located centrally and 55 peripherally.

Fourteen further cases with carcinoma on the left and twelve cases on the right side presented with tumor manifestation in the main bronchus and could not be assigned to any lung lobe. In addition, in three cases, the tumor is located centrally supracarinal and could not be assigned to a lung lobe.

Regarding lung metastases, 16 cases of metastases are located in the left lung and 33 metastases are located in the right lung. In one case, the tumor is located very centrally and could not be assigned to any lung lobe. In the left lung, six cases are located in the central segments of the lung lobes, and nine cases are in the peripheral lung lobes. In the right lung, 15 cases are located centrally and 14 peripherally. Information regarding the tumor location for cases presenting with metastasis can be found in Table 3.

In one further case with a tumor on the left side and four further cases with a tumor on the right side, the lesion is located in the main bronchus and could not be assigned to any lung lobe. The same applies to one case with a supracarinal tumor.

### 3.4. Histological Diagnosis of Lung Carcinoma in Small Biopsies According to the 5th WHO Classification

Of the 242 patients with primary lung carcinoma, a significant number (112 (46.3%)) could be diagnosed directly on histomorphology alone. Sixty-one cases (54.5%) had a classical morphology of an ADC, and fifty-one (45.6%) had a classical morphology of a SCC (Figure 1A,B).

An additional number of 19 cases (7.8%) could be diagnosed with the use of a PAS staining for the recognition of mucin (in 18 cases, Figure 1C,D) or the distinction between acantholysis or acinar differentiation (in 1 case). The remaining 111 cases (45.9%) required immunohistochemical techniques.

In 37 cases (15.3%), there was a morphological suspicion of a small-cell lung carcinoma (SCLC), in 2 cases (0.8%) of typical carcinoid, in 2 (0.8%) cases of atypical carcinoid tumor, and in 1 (0.4%) case of a large-cell neuroendocrine carcinoma (LCNEC), so they were subjected to immunohistochemical investigation for synaptophysin (Figure 2).

In all four cases with carcinoid tumors, we were able to subclassify them into typical and atypical on the grounds of the histomorphology alone, using the criteria of necrosis, cellular atypia, and number of mitoses. Among the cases with the suspicion of SCLC, in 12 of them, we observed some scattered cells with light-enlarged nuclei. This finding was associated with uncertainty concerning the distinction between small- and large-cell neuroendocrine carcinoma in these cases, but the diagnosis of SCLC was made according to the guidelines of the recent WHO classification [2]. These cases were also subjected to further analysis (see Section 3.9). In one additional case, the distinction from a basaloid squamous cell carcinoma was not possible on morphological grounds, and subsequent additional analyses were performed. In two further cases, a combined SCC with LCNEC was diagnosed, using only synaptophysin staining for the confirmation of neuroendocrine differentiation in the second tumor component.

In the remaining cases, we performed additional ancillary techniques. Among these cases, 31 were diagnosed as ADC on the basis of TTF-1 positivity and 12 as SCC due to p40 positivity. In three more cases, the diagnosis of a LCNEC was made based on the combination of morphology with the expression of synaptophysin and the absence of TTF-1 or p40 expression. In the remaining 19 cases (7.8%), the diagnosis of a NSCLC NOS was made. Table 4 summarizes the number of ancillary techniques made in cases of each histological type of lung carcinoma. The diagnostic procedure used during our evaluation is illustrated in Figure 3.

Cases diagnosed on bronchoscopy-gained material need a mean number of 1.3 ancillary techniques, and 38.6% of the cases could be subclassified without any additional testing. For EBUS-TBNA-based diagnoses, a mean number of 1.2 additional stains is necessary, and 42% of the cases could be classified based on morphology alone.

For biopsies gained via CT- or ultrasound-guided transthoracal puncture, a mean number of 0.44 (CT-guided) and 1.15 (ultrasound-guided) additional techniques are necessary. Moreover, 77.8% of CT-guided) and 42.4% of the ultrasound-guided cases were diagnosed by morphological presentation alone.

### 3.5. Comparison of the Application of the Criteria of 5th WHO Classification with the Extensive Use of Ancillary Techniques

The mean number of ancillary techniques used during the application of the criteria for diagnosing lung cancer in small biopsies as suggested by the recent WHO classification [2] is 1 (range 0–4, SD ± 1.4). On the contrary, the mean number of ancillary techniques used for the diagnosis according to our archive was significantly higher, namely six (range 0–15, SD ± 3), and this difference was proven to be statistically significant (paired *t*-test, *p* < 0.001). We were able to make the same diagnosis using a limited immunohistochemical panel with the diagnosis made with the extensive use of immunohistochemical markers in the vast majority of the examined cases, namely 97.3% of the cases (concordance rate 237/242, paired *t*-test, *p* > 0.10). In three cases, the original diagnosis was squamous carcinoma due to cytokeratin 5 immunohistochemical expression, which we were not able to recognize with the limited use of immunohistochemistry and thus made a diagnosis of a NSCLC NOS (Figure 4A–C). In one case, we did not have a morphological suspicion of neuroendocrine morphology and, without performing respective immunohistochemical analysis, we made the diagnosis of a NSCLC NOS, but archival immunohistochemistry revealed an extensive expression of synaptophysin and chromogranin, leading to the diagnosis of a LCNEC (Figure 4D,E). Finally, there was a case in which we made the diagnosis of a SCC morphologically due to the presence of individual cell keratinization, but originally, the diagnosis of a NSLC NOS was made. In four of these cases, the diagnosis was made on bronchoscopically obtained material, and in one, on CT-guided puncture material.

### 3.6. Histological Diagnosis of Cases with Metastasis in Small Lung Biopsies

Clinical information for the suspicion of metastatic disease due to clinical history or clinical presentation (for example multiple lung lesions) was shared with us according to our records in 29 cases (61.7%), among the 47 lung biopsies with a metastasis. In four more additional cases (8.5%), we had a previous histological report of the patient in our system with a diagnosis of a malignant tumor, and the question of a metastasis was made by the pathologist due to this information. In the remaining 14 cases (29.8%), there was no available information regarding the suspicion of a metastatic disease. In the cases with suspicion of metastasis, we were able to reach the correct diagnosis with a mean number of two ancillary stainings (0–5, SD ± 1.6). It is important to note that 14 cases could be diagnosed without performing any immunohistochemistry.

Regarding the 14 cases without a suspicion of a metastasis, in 5 of them, we were able to make the diagnosis of metastasis on morphological grounds (1 melanoma, 1 colorectal carcinoma, 1 breast carcinoma, 1 thyroid carcinoma, and 1 SCC). In the remaining nine cases, we would have made the diagnosis of a NSCLC NOS if we had performed only a limited number of immunohistochemical stains, as suggested by the latest WHO classification (3). These nine cases consisted of four renal cell carcinomas, one breast carcinoma, one prostate carcinoma, two melanomas, and one SCC. The mean number of immunohistochemical techniques originally performed in these nine cases in order to reach the correct diagnosis was 11 (range 6–15, SD ± 4.3).

An investigation of our ability to make the correct diagnosis in the whole group of 47 lung biopsies with metastasis in case there was not any suspicion of a metastatic disease (shared by the clinicians or according to our records) was undertaken. In the absence of this suspicion, we were able to make the correct diagnosis in 44.7% of the cases (21/47) based on histomorphology. In the rest of them (26/46, 55.3%), morphology or limited immunohistochemical analysis was not sufficient for the correct diagnosis. Figure 5 illustrates some examples of cases in which the morphology was not sufficient for the diagnosis. Based on our records, the mean number of immunohistochemical stains performed originally in these cases to reach the final diagnosis was 10 (range 6–20, SD ± 3.9). Table 5 presents the correct final diagnosis of these 26 cases.

### 3.7. Comparison of the Diagnosis Made in Small Biopsies with the Complete Excision

In 39 cases, we received a lung specimen with a corresponding lesion to the previous lung biopsy in our department. In 31 of these cases, the diagnosis made on biopsy was confirmed (concordance between diagnoses 79.5% of the cases, paired *t*-test, *p* > 0.10). In two cases, there was not any residual tumor present after neoadjuvant therapy (ypT0). Four cases diagnosed as NSCLC NOS in bioptical material were further classified into either an ADC (two cases), a SCC (one case), or an adenosquamous carcinoma (one case). One case diagnosed as atypical carcinoid in the biopsy was diagnosed as LCNEC in the large specimen. Finally, one case diagnosed as SCC in the biopsy was classified into the group of large-cell lung carcinomas in the lung specimen.

### 3.8. Comparison of p40 with p63

There was a statistically significant difference between the expression levels of p63 and p40 (Student *t*-test, *p* < 0.001, Chi-square test < 0.001). In particular, 53% of the cases were negative for p40, 10.6% showed a minimal expression (<10%), 4.6% a moderate expression (10–50%), and 31.8% an extensive expression (>50%). On the other hand, 39.7% of the cases were negative for p63, 15.6% showed a minimal expression (<10%), 7.5% a moderate expression (10–50%), and 37.2% an extensive expression (>50%). Importantly, among the 104 p40-negative cases, 32 displayed a positive p63 immunoreactivity (30.7%), 22 showed a minimal expression (<10%), 7 a moderate expression (10–50%), and 3 an extensive expression (>50%) (Figure 6).

No difference was observed in the level of p40 and p63 expression for the SCC group (Student *t*-test, *p* > 0.10), but the difference was found to be statistically significant for the subgroup of ADC (Student *t*-test, *p* < 0.001) or NSCLC NOS (Student *t*-test, *p* = 0.007), with p63 having the tendency to display higher levels of immunohistochemical expression compared with p40. However, in none of the investigated cases would this difference be able to change the final histological diagnosis. In the absence of p40 immunoreactivity, p63 expression was either accompanied by mucin production or TTF-1 expression, leading to the correct diagnosis of ADC, or it was not so extensive (<50%), thus leading to the diagnosis of NSCLC NOS.

### 3.9. p53 and Rb1 Expression in Neuroendocrine Carcinomas

As previously mentioned, in 12 cases of the SCLC cases, there was a certain uncertainty regarding the distinction from LCNEC due to the presence of some scattered cells with light-enlarged nuclei, and in one case, the distinction from basaloid SCC could not be made on a morphological basis. In these 13 cases, we decided to perform a subsequent immunohistochemical analysis with p53 and Rb1. None of these cases showed a nuclear expression of Rb1. Additionally, nine cases (9/13, 69.2%) displayed either no expression (null phenotype, 30.8%) or an aberrant expression of p53 (38.4%), indicating the presence of a p53 mutation (Figure 7).

We also tested three LCNECs for the expression of p53 and Rb1. Interestingly, in one case, we observed a nuclear overexpression of Rb1 and p53, with the other two not showing any p53 (null phenotype) or nuclear Rb1 immunoreactivity.

### 3.10. PD-L1 Expression in Lung Carcinomas

Immunohistochemical analysis of PD-L1 expression was available in 197 cases with primary lung cancer. In all these cases, we were able to evaluate TPS, but in eight cases, the evaluation of IC-score was not possible due to the absence of tumor-infiltrating inflammatory cells. This was mainly observed in EBUS-TBNA material (six cases) and less in bronchial biopsy (two cases).

TPS was positive (>1%) in 116 of the examined cases (58.9%): 58.7% of ADCs, 61.9% of SCCs, and 66.6% of NSCLC NOS. IC-score was 3 in 73 of the available cases (38.6%): 28.6% of ADCs, 41.7% of SCCs, and 77.8% of NSCLC NOS. Among the cases without any PD-L1 expression in the tumor cells, 24 displayed a negative IC-score (30.8%), 17 had an IC-score of 1 (21.8%), 10 had an IC-score of 2 (12.8%), and 27 had an IC-score of 3 (34.6%). The cases with a negative TPS and an IC-score of 3 had the following distribution among histological types: 12 ADCs, 8 SCCs, 4 NSCLCs, and 3 LCNECs (Figure 8).

### 3.11. Molecular Analysis

Forty-five of the lung carcinoma cases were subsequently submitted to molecular analysis which, despite the extensive immunohistochemical panel performed, it was possible to be conducted. Twenty-eight cases had a K-RAS mutation (62.2%), eight a EGFR mutation (17.8%), three a RET Fusion (6.7%), two a MET Fusion (4.4%), one a ROS1 Fusion (2.2%), one a CTNNB-1 mutation (2.2%), and one an inactivating B-RAF mutation (2.2%).

## 4. Discussion

For several decades, the lack of specific targeted therapy among different types of lung cancer negated the necessity for sophisticated pathological diagnosis. During the last two decades, our understanding of lung cancer biology has led to fast development, emerging into the era of precision medicine. The relevant reported improvements in patient outcomes are testaments in this direction [9]. These recent advances urge a precise subclassification of NSCLC into specific histological subtypes with the use of immunohistochemistry and the following testing of predictive biomarkers [10].

From the histopathological point of view, there have been several improvements in the classification of lung cancer, with the description also of rare subtypes, such as SMARCA4-deficient undifferentiated tumors [2,9]. The recent 2021 WHO classification of thoracic tumors emphasizes the role of the pathologist in small lung biopsies, which represent the vast majority of the specimens received in the departments of pathology worldwide nowadays, in a whole separate section, dedicated to the classification and the nomenclature of small diagnostic samples [2,10]. This section describes the workflow for managing small biopsy and cytology samples and emphasizes the necessity for an accurate histological diagnosis [2]. According to the described workflow for managing small biopsies and cytological specimens, adequate histological diagnosis of lung cancer in small biopsies should be made, when possible, by morphology alone, and it requires a minimum panel of immunohistochemical studies in an effort to spare as much tissue for molecular testing. These recommendations, however, increase the challenges that the pathologist has to encounter, from limited diagnostic material to difficulties in histological interpretation, with a concurrent effort to avoid extended investigations and at the same time secure an accurate diagnosis [5].

In our study, we made an effort to use the diagnostic workflow suggested in the latest 2021 WHO classification of thoracic tumors in a series of 288 patients with small lung biopsies and tried to compare the results with the diagnosis made in our system based on an extensive use of immunohistochemical techniques. We found that the use of WHO-proposed criteria for the diagnosis of lung carcinoma based on small lung biopsies is associated with high diagnostic certainty, with a concordance rate of more than 97% in the subgroup of primary lung carcinomas. Importantly, a great number of cases (112 (46.3%)) could be diagnosed only on morphological grounds, without using any additional techniques. This result is important, taking into consideration that the mean number of ancillary techniques with this workflow is 1 (SD ± 1.4), compared with a mean number of 6 (SD ± 3) additional tests according to our archive. As expected, the number of additional stains depended largely on the type and quantity of the available biopsy material, since CT-guided biopsies, usually yielding more material and often having a well-preserved morphology, required fewer stains compared with the smaller tissue samples (often obtained by EBUS-TBNA or transbronchial lung biopsy). This diagnostic approach can be easily integrated into the daily routine and warrants awareness from the practicing pathologist to perform fewer additional techniques for this purpose in order to save time and costs, as well as tissue for further molecular analysis. According to our experience, 10–15 additional sections are necessary in each case in order to perform additional testing, such as fluorescent in situ hybridization (FISH) or new generation sequencing (NGS) in order to be able to follow the guidelines for the companion diagnostic. Currently, the NCCN guidelines strongly advise testing NSCLC for epidermal growth factor (EGFR), anaplastic lymphoma kinase (ALK), ROS proto-oncogene 1 (ROS), rearranged during transfection proto-oncogene (RET), MET proto-oncogene (MET) exon 14 skipping, v-raf murine sarcoma viral oncogene homolog B1 (BRAF), and neurotropic tyrosine kinase receptor kinase (NTRK) mutations, all of which have approved targeted therapies [5]. In the same context, MET amplification, KRAS G12C, and ERBB2 (HER2) mutations are also suggested as emerging biomarkers [5].

The subgroup of five cases with a divergent diagnosis consists of three cases where a squamous differentiation could only be recognized through a CK5-stain, which is not part of the WHO-proposed panel. During this workflow, these three cases were classified into the group of NSCLC NOS, since they also did not show the classical features of squamous differentiation in the form of keratin pearls, individual cell keratinization, and/or intercellular bridges. According to the respective literature, p40 immunohistochemical positivity has a high sensitivity in the detection of squamous differentiation [11,12]. There are, however, several reports showing a sensitivity rate of approximately 77% [13]. The rare incidence of a p40-negative squamous cell carcinoma is well known among pathologists, so several institutes have introduced dual staining for p40/CK5 in order to be able to recognize the p40-negative nonkeratinizing SCC [5]. However, this dual staining procedure is not available in many institutes, mainly due to economic barriers. In the same context, CK5 staining has often been suggested in the literature as an additional marker in rare indeterminate cases [14] and should be kept in mind in some cases of NSCLC NOS.

Another case with a divergent diagnosis in our cohort was a case in which, due to the absence of neuroendocrine morphology, no immunohistochemical investigation for synaptophysin was performed, and therefore, by the negativity of TTF-1/p40 and no conclusive morphology, a NSCLC NOS was diagnosed. This case was diffusely positive for synaptophysin and therefore diagnosed as a LCNEC according to our archive. However, as already mentioned, the recent WHO classification of thoracic tumors recommends not diagnosing a neuroendocrine carcinoma and not performing the respective immunohistochemical analysis in the absence of neuroendocrine morphology [2]. Several reports in the literature have proposed algorithms for the detection of LCNEC in small lung biopsies, including, however, an increased number of immunohistochemical investigations, such as synaptophysin, chromogranin, and/or Ki67 for this purpose [15,16]. There remains, however, the question of whether our case really represents a LCNEC or not, and if this question has a clinical relevance, since there has been growing evidence that platinum/etoposide-based regimens utilized for SCLC show similar efficacy in patients with LCNEC and are superior to NSCLC treatments in the adjuvant setting and as first-line therapy for advanced disease [16,17,18,19].

The last case with a divergent diagnosis showed individual cell keratinization, which was not recognized in our initial diagnostic procedure and, according to the subsequent immunohistochemical negativity for TTF-1 and p40, was classified as a NSCLC NOS. This case emphasizes the importance of morphology in the diagnosis of lung carcinoma. Pathologists should always bear in mind that the diagnosis of lung cancer in small biopsies is by definition a morphological-based procedure, and morphology remains a not easily replaceable tool for lung cancer classification [2,20]. In this context, it is also important to have a consultation with an experienced pathologist in these inconclusive cases before the diagnosis of a NSCLC NOS is made. There are currently several attempts to develop deep learning-based techniques for the diagnosis and subclassification of these morphologically and immunohistochemically challenging and indeterminate cases, which have shown preliminary good results [21]. Digital pathology could also play a significant role in scant diagnostic samples. Further work in this field is, however, necessary before it can be implemented in the daily diagnostic routine.

In our investigation, we were also able to compare our diagnosis based on small biopsies with the diagnosis made in subsequent lung specimens in 39 of the cases. As expected, we were able to further subclassify cases diagnosed as NSCLC NOS in small biopsies in specific histological types, namely ADC, a SCC, or an adenosquamous carcinoma. One case diagnosed in the biopsy as an atypical carcinoid was diagnosed as LCNEC in the large specimen, which could possibly be attributed to a sample error caused by tumor heterogeneity. In general, the concordance rate between the two diagnoses was relatively high (almost 80%).

However, a different situation emerges when looking at the cases with metastases in the lung from a primary tumor located elsewhere in the body. The WHO classification of thoracic tumors does not recommend excluding possible lung metastasis by immunohistochemical testing of all transcription factors [2] and emphasizes the role of the clinician in this purpose. Interestingly, in almost 30% of the metastatic cases, we did not have any previous records either from our system or from the provided clinical information regarding the possibility of metastasis. We found that in the absence of clinical information, only 44.7% of the cases would have been correctly diagnosed as metastases merely based on morphological assessment and limited immunohistochemical panel. From the remaining cases, 18 cases would have been misdiagnosed as NSCLC NOS and 8 cases as primary adenocarcinoma of the lung. However, if sufficient clinical information was available, all metastases could be successfully recognized. In addition, a significantly smaller number of additional examinations (mean of 2 vs 10) were necessary in this situation, which could also play a role in sparing tissue for molecular analysis in these cases. These results emphasize the role of clinical information, as well as adequate communication between the pulmonologist and the treating oncologist to ensure a sufficient multidisciplinary approach and avoid overuse and wastage of tissue from the side of the pathologist [22].

Our study compared the results of p40 and p63, the two available immunohistochemical markers for the recognition of squamous differentiation. Until recently, p63 was the most frequently used marker for this purpose, but it has been reportedly associated with a relatively low specificity, being also positive in 15–65% of ADC and some NSCLC NOS cases [23,24,25]. In alignment with these observations, we also found a significant difference in the expression levels of these markers in ADC and NSCLC, with p63 tending to be more often present and expressed in higher levels than p40. However, the increased expression of p63 compared with p40 was not sufficient to influence the final diagnosis. These results imply that although p63 seems to have a lower specificity compared with p40, it can also be used for this diagnostic purpose, always keeping in mind, however, that SCC shows an extensive p63 expression and that p63 staining should be interpreted together with other stains (TTF-1 or mucin) for the final diagnosis.

Furthermore, in our investigation, we found that several SCLCs displayed some scattered cells with enlarged nuclei and these cases were associated with uncertainty concerning the distinction between SCLC and LCNEC. The recent WHO classification of thoracic tumors accepts the presence of some enlarged cells in SCLC cases [2], and therefore, we made this final diagnosis. In an attempt to investigate if the expression of p53 and Rb1 could be helpful for this purpose, we found that all these cases had a combination of cytoplasmic Rb1 with aberrant/no expression of p53. Recent data support the notion that the inactivation of RB and p53 represent the initial steps in the development of SCLC, making them essential for a lung epithelial cell to progress toward the acquisition of a malignant phenotype [2,26]. On the other hand, some LCNECs showed the same phenotype regarding Rb1 and p53, which was expected since, according to the literature, a subgroup of LCNECs have TP53/RB1 co-alterations and are SCLC-like [2,27,28]. Our observations suggest that the absence of p53/Rb1 co-alterations could be used as an additional diagnostic tool in cases where this diagnosis is doubted due to the presence of some enlarged nuclei.

Another interesting finding of the present investigation is associated with PD-L1 expression in lung biopsies. The current guidelines for lung carcinoma propose the analysis of PD-L1 expression in every case of NSCLC, using a validated PD-L1 immunohistochemistry expression assay to optimize selection for treatment with immune checkpoint inhibitors [28]. Our purpose was to see how many patients without a positive PD-L1 TPS displayed an increased IC-score, thus having the opportunity to profit from therapy with a specific immune checkpoint inhibitor, atezolizumab, which seems to have an efficacy in patients with PD-L1 TPC ≥ 50% or IC ≥ 10% [29]. According to the literature, TPS is less than 1% in 41–57% of patients with NSCLC [30,31]. In general, it remains unclear whether patients with negative PD-L1 expression can benefit from immunotherapy [30,31]. In our data, there are 27 cases with a negative TPS and an IC-score of 3, composed of 12 ADCs, 8 SCCs, 4 NSCLCs, and 3 LCNECs. Several studies are currently being carried out with regard to this issue and with regard to a response to therapy in PD-L1-negative tumors according to TPS score [32]. In this context, there are reports of better efficacy of some immuno-combined chemotherapy regimens than chemotherapy alone in patients with negative PD-L1 TPS expression [30]. Apart from PD-L1, there are currently several additional biomarkers that are being investigated for their potential role in tumor differentiation, proliferation, or aggressiveness in lung cancer and could in the future be integrated into the daily routine as prognostic and/or predictive biomarkers [33]. In this context, for example, p16 expression has been recently associated with a worse prognosis in early-stage diseases [34].

## 5. Conclusions

The diagnostic workflow suggested by the recent WHO classification of thoracic tumors for small lung biopsies can lead to high diagnostic certainty and save material for further molecular testing, as well as time, effort, and costs. However, lack of clinical information and inadequate communication between the treating clinician and the pathologist may result either in tissue overusing from the pathologist due to diagnostic uncertainty or may increase the possibility of failing to distinguish a metastatic disease from a primary tumor. Under these circumstances, the diagnosis of a NSCLC NOS is justifiably accompanied by diagnostic uncertainty, and the pathologist should in this case be alerted of the possibility of trying to recruit clinical information or using a more extensive panel (for example, including CK5 for the case of p40 negative SCC or other transcription factors to exclude metastasis). Moreover, according to our investigation, although p63 has a lower specificity in the diagnosis of SCC and is more highly expressed in other histologic types compared with p63, it remains a reliable marker for this purpose. Furthermore, our study suggests that the combined investigation of Rb1/p53 expression can be useful in the exclusion of the diagnosis of SCLC in cases with a controversial morphology. A combined evaluation of both TPS and IC-score is necessary and could provide us with more information regarding the role of immunotherapy in PD-L1 TPS-negative cases.

## Figures and Tables

**Figure 1 diagnostics-14-02090-f001:**
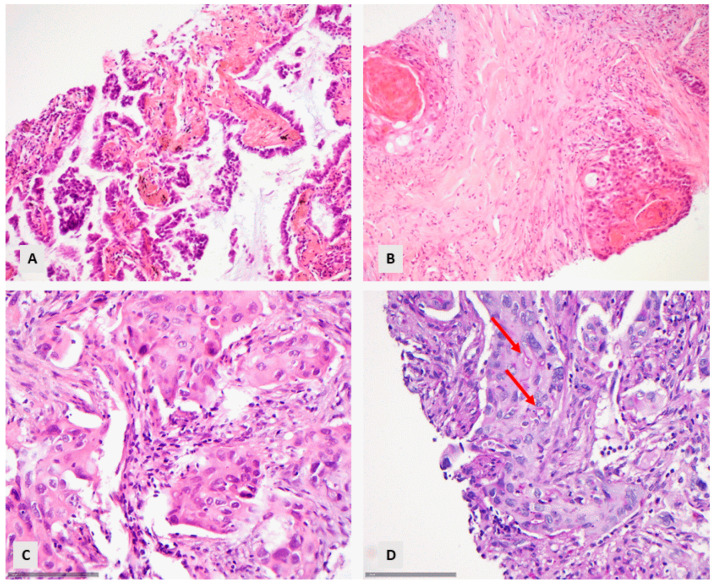
Cases diagnosed on morphological grounds: (**A**) Lung adenocarcinoma with lepidic morphology (HE, ×160). (**B**) Squamous cell carcinoma with keratinization (HE, ×160). (**C**,**D**) Lung adenocarcinoma with PAS-positive mucin ((**C**) HE, ×400, (**D**) PAS, ×400). The arrows illustrate mucin-containing intracytoplasmic vacuoles.

**Figure 2 diagnostics-14-02090-f002:**
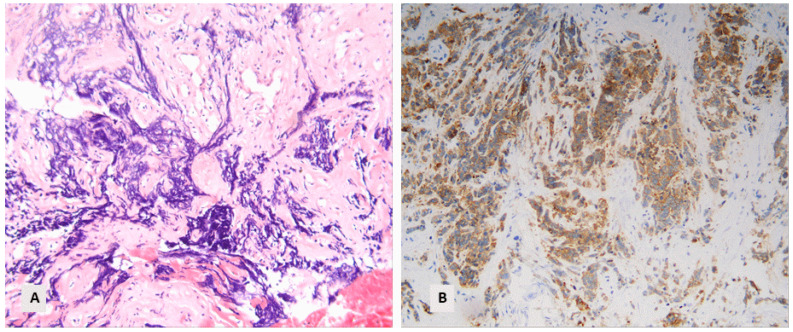
A small-cell lung carcinoma with positive reaction for synaptophysin: (**A**) Characteristic morphology with small cells showing crush artifacts (HE, ×160). (**B**) Positive immunohistochemical stain for synaptophysin (×160).

**Figure 3 diagnostics-14-02090-f003:**
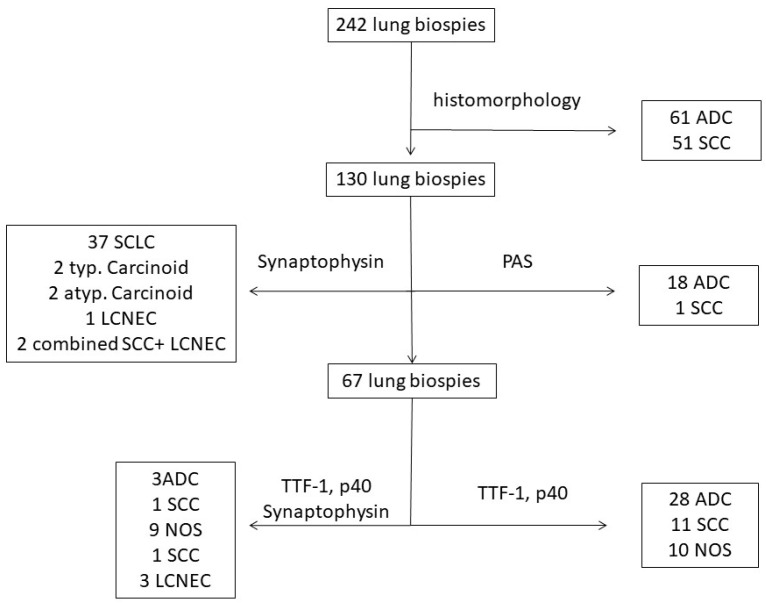
Diagnostic procedure for the diagnosis of lung carcinoma in small biopsies according to the guidelines of the 5th WHO classification of lung cancer.

**Figure 4 diagnostics-14-02090-f004:**
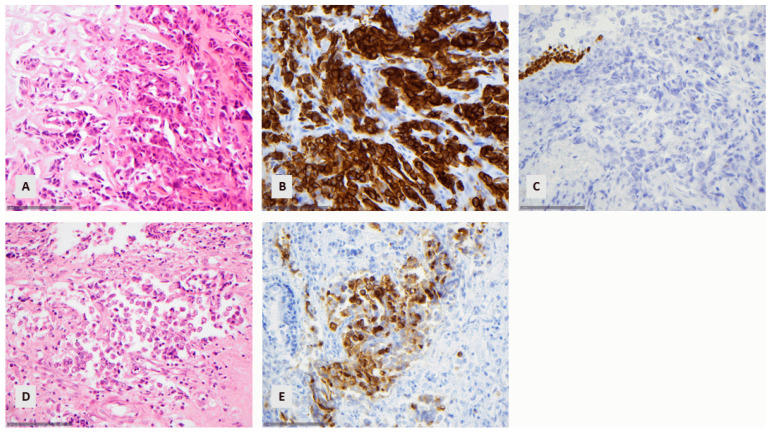
(**A**–**C**) A squamous cell carcinoma being negative for p40 and positive for CK5: (**A**) The neoplastic cells are medium-sized and there is no keratinization (HE, ×400). (**B**) Positive reaction for CK5 (×400). (**C**) Negative reaction for p40 (×400). The positive reaction of the superficial epithelium is noted. (**D**,**E**) NSCLC with positive synaptophysin. (**D**) The tumor cells are medium to large with vesicular nuclei and abundant cytoplasm, favoring the diagnosis of a NSCLC (×400). (**E**) Positive reaction for synaptophysin (×400).

**Figure 5 diagnostics-14-02090-f005:**
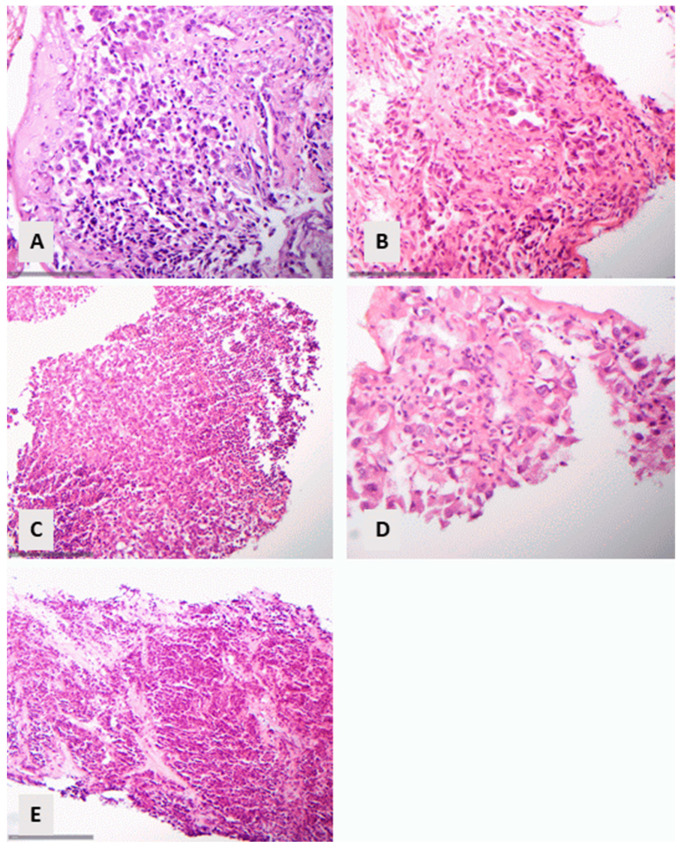
Metastatic cases: (**A**) breast cancer NST (HE, 400). (**B**) breast cancer NST (HE, ×400). (**C**) renal cell carcinoma (HE, ×200). (**D**) renal cell carcinoma (HE, ×400). (**E**) prostate cancer (HE, ×200).

**Figure 6 diagnostics-14-02090-f006:**
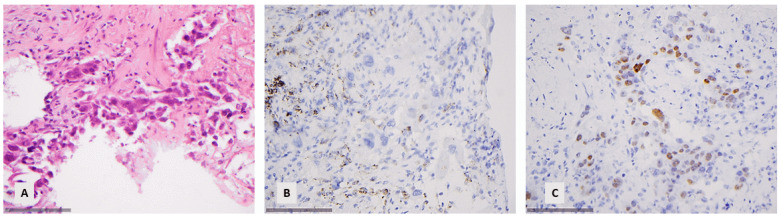
A NSCLC with extensive expression of p63 and negativity for p40 ((**A**) HE, ×400, (**B**) p40, ×400, (**C**) p63, ×200).

**Figure 7 diagnostics-14-02090-f007:**
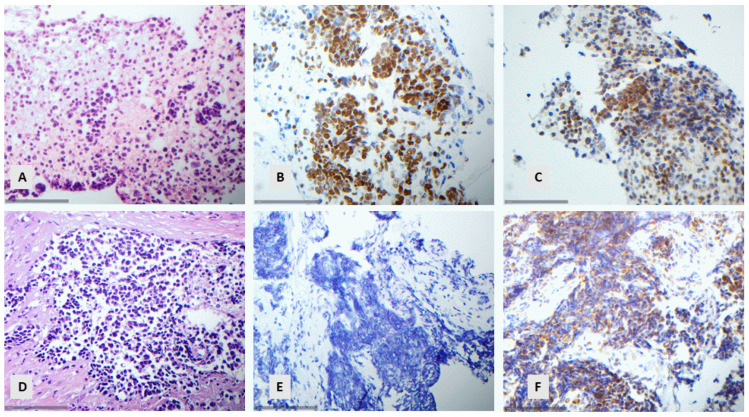
(**A**–**C**) SCLC with aberrant expression of p53 and cytoplasmic Rb1 ((**A**) HE, ×400. (**B**) p53. ×400. (**C**) Rb1. ×400). (**D**–**F**) SCLC with null expression of p53 and cytoplasmic Rb1 ((**D**) HE, ×400. (**E**) p53. ×400. (**F**) Rb1. ×400).

**Figure 8 diagnostics-14-02090-f008:**
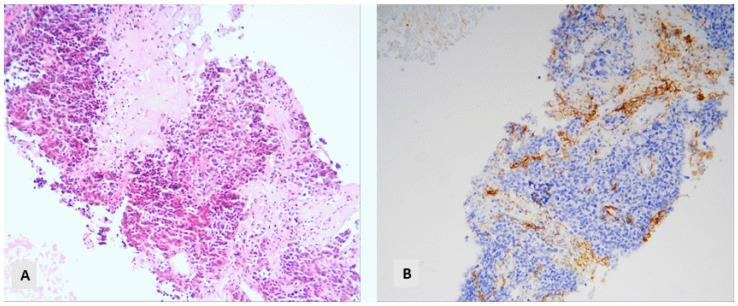
NSCLC with a negative TPS score and a positive IC-score: (**A**) HE, ×160. (**B**) PD-L1 stain (×160).

**Table 1 diagnostics-14-02090-t001:** Patients’ demographic characteristics.

	Entire Cohort	Lung Carcinomas	Metastasis
	Mean (range, SD)	Mean (range, SD)	Mean (range, SD)
Age (years)	68 (50–90, SD ± 9.9)	69 (41–90, SD ± 9.7)	69 (34–95, SD ± 10.6)
	Male/Female	Male/Female	Male/Female
Gender	175/113	146/96	29/17

**Table 2 diagnostics-14-02090-t002:** Tumor location of primary lung carcinomas.

	Upper Lobe (Left)	Lower Lobe (Left)	Upper Lobe (Right)	Middle Lobe (Right)	Lower Lobe (Right)	Total
Central	45	22	27	6	22	122
Periphery	22	14	28	6	21	91
Total	67	36	55	12	43	

**Table 3 diagnostics-14-02090-t003:** Tumor location of lung metastases.

	Upper Lobe (Left)	Lower Lobe (Left)	Upper Lobe (Right)	Middle Lobe (Right)	Lower Lobe (Right)	Total
Central	4	2	7	6	2	21
Periphery	3	6	9	1	4	23
Total	7	8	16	7	6	

**Table 4 diagnostics-14-02090-t004:** Distribution of the number of ancillary techniques performed among the different histological types of lung cancer.

	Number of Cases			
Histological Type	No Ancillary Techniques	1	3	4
ADC	61	18	28	3
SCC	52	1	11	1
SLCL	38	-	-	1
LCNEC	-	1	3	-
Typical carcinoid	2	-	-	-
Atypical carcinoid	2			
NSCLC NOS	-	-	10	9
Combined SCC + LCNEC	2			

**Table 5 diagnostics-14-02090-t005:** Cases in which morphology and/or limited immunohistochemical analysis in the absence of suspicion of metastasis (according to provided clinical information or to our records) were not sufficient for the correct diagnosis.

Diagnosis	Number of Cases	%
Cholangiocarcinoma	1	3.9
Colorectal Ca	2	7.7
Endometrial Ca	2	7.7
Lung Ca	1	3.9
Breast Ca	4	15.4
Melanoma	3	11.5
Renal cell Ca	7	26.9
Prostate Ca	2	7.7
Upper GI Ca	1	3.9
Urothelial Ca	2	7.7
SCC (from Head and neck region)	1	3.9

Ca: Carcinoma.

## Data Availability

The data presented in this review are available on request from the corresponding authors.

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
