# Peer review of "Application of the 5th WHO Guidelines for the Diagnosis of Lung Carcinoma in Small Lung Biopsies in a Tertiary Care Center: Is Insecurity of Pathologists for the Accurate Diagnosis Justified?"

_diagnostics, 2024, doi:10.3390/diagnostics14182090_

Round 1
Reviewer 1 Report
Comments and Suggestions for Authors
The manuscript is technically correct. In my opinion, few minro suggestions shoudl be implemented to accept this paper for the publication.
- Please, could the authors show how this approach may be implemented in clinical practice?
- Please, could the authros discuss about digital pathology to solve challenging issues?
- Please, could the authors discuss about the scant diagnostic sampels routinely available to be diagnosed?
Comments on the Quality of English Languageminor english editing
Author Response
Thank you for your interest in our article and your valuable suggestions.
We modified our manuscript according to your suggestions, as follows:
- Please, could the authors show how this approach may be implemented in clinical practice?
We commented upon the implementation of this approach into the clinical practice, which in our opinion warrants only awareness from the practicing pathologist that this approach is efficient and saves time, costs and tissue.
- Please, could the authors discuss about digital pathology to solve challenging issues?
The potential role of digital pathology in challenging and indeterminate cases has been included in the discussion section and we added a recent literature on this field (Karavati et al, 2021).
- Please, could the authors discuss about the scant diagnostic samples routinely available to be diagnosed?
The difficulty in diagnosing scant diagnostic samples has been now included in the discussion section of the revised version of our manuscript, by emphasizing our result that the small samples required a higher number of additional techniques.
-minor English editing
The manuscript has been reviewed from a native English speaker for potential errors and accordingly corrected.
Reviewer 2 Report
Comments and Suggestions for Authors Additional explanation about statistical test choises should be providen Why did you use t-test in some casese and Wilcoxon rank test in others? please include data regarding smoking habit. I suggest to discuss about other histochemical markers of differentiation, proliferation and aggressiveness I also suggest to include the following reference for the discussion Anticancer Res. 2020 Feb;40(2):983-990.Author Response
Thank you for your interest in our article and your valuable suggestions.
We modified our manuscript according to your suggestions, as follows:
-Additional explanation about statistical test choises should be providen Why did you use t-test in some casese and Wilcoxon rank test in others?
Thank you for your valuable comment. Since we found a normal distribution of our data, according to Kolmogorov-Smirnov test, we now used only paired T-test for our analysis and Wilcoxon paired test has been removed.
-please include data regarding smoking habit.
Unfortunately, information regarding smoking habit is not available in our cohort.
-I suggest to discuss about other histochemical markers of differentiation, proliferation and aggressiveness I also suggest to include the following reference for the discussion Anticancer Res. 2020 Feb;40(2):983-990.
The potential role of additional markers of differentiation, proliferation or aggressiveness in lung cancer and their implementation has been now included in the discussion section of our manuscript. The suggested reference has also been added.